# The Effect of Different Physical Exercise Programs on Physical Fitness among Preschool Children: A Cluster-Randomized Controlled Trial

**DOI:** 10.3390/ijerph20054254

**Published:** 2023-02-27

**Authors:** Guangxu Wang, Dan Zeng, Shikun Zhang, Yingying Hao, Danqing Zhang, Yang Liu

**Affiliations:** 1School of Physical Education, Shanghai University of Sport, Shanghai 200438, China; 2College of Physical Education, Henan Normal University, Xinxiang 453007, China; 3National Institute of Sports Medicine, Beijing Sport University, Beijing 100084, China; 4School of Exercise and Health, Shanghai University of Sport, Shanghai 200438, China; 5Shanghai Research Center for Physical Fitness and Health of Children and Adolescents, Shanghai University of Sport, Shanghai 200438, China

**Keywords:** physical exercise, basic movements, rhythm activities, ball games, multiple activities, physical fitness, preschool children

## Abstract

Background: Preschool children are in a period of rapid physical and psychological development, and improving their level of physical fitness is important for their health. To better develop the physical fitness of preschool children, it is very important to understand the behavioral attributes that promote the physical fitness of preschool children. This study aimed to determine the effectiveness of and the differences between different physical exercise programs in improving preschool children’s physical fitness. Methods: A total of 309 preschool children aged 4–5 years were recruited from 5 kindergartens to participate in the experiment. They were cluster-randomly allocated into five groups: basic movements (BM) group, rhythm activities (RA) group, ball games (BG) group, multiple activities (MA) group, and control (CG) group. The intervention groups received designed physical exercise programs with a duration of 30 min 3 times per week for 16 weeks. The CG group received unorganized physical activity (PA) with no interventions. The physical fitness of preschool children was measured using the PREFIT battery before and after the interventions. One-way analysis of variance, a nonparametric test; generalized linear models (GLM); and generalized linear mixed models (GLMM) were used to examine differences during the pre-experimental stage among groups and to assess the differential effects of the intervention conditions on all outcome indicators. The intervention condition models were adjusted for potential confounders (baseline test results, age, gender, height, weight, and body mass index) explaining the main outcome variance. Results: The final sample consisted of 253 participants (girls 46.3%) with an average age of 4.55 ± 0.28 years: the BG group (n = 55), the RA group (n = 52), the BM group (n = 45), the MA group (n = 44), and the CG group (n = 57). The results of the generalized linear mixed model and generalized linear model analyses indicated significant differences for all physical fitness tests between groups, except for the 20 m shuttle run test and the sit-and-reach test after the interventions. Grip strength was significantly higher in the BG and MA groups than in the BM group. The scores for standing long jump were significantly higher in the MA group than in the other groups. The scores for the 10 m shuttle run test were significantly lower in the BG and MA groups than in the CG, BM, and RA groups. The scores for skip jump were significantly lower in the BG and MA groups than in the RA group. The scores for balance beam were significantly lower in the BG and MA groups than in the RA group and significantly lower in the BG group than in the BM group. The scores for standing on one foot were significantly higher in the BG and MA groups than in the CG and RA groups and significantly higher in the BM group than in the CG group. Conclusions: Physical exercise programs designed for preschool physical education have positive effects on the physical fitness of preschool children. Compared with the exercise programs with a single project and action form, the comprehensive exercise programs with multiple action forms can better improve the physical fitness of preschool children.

## 1. Introduction

Physical fitness can be considered as the comprehensive performance of physical functions, such as muscular function, cardiovascular function, and metabolic function, effectively during daily physical activity (PA) or physical exercise [1]. Healthy levels of physical fitness guarantee that individuals participate in physical activity and work with vigor, and can promote resistance to fatigue [2]. Studies have indicated that a series of health problems in children are related to low levels of cardiorespiratory fitness and muscle strength, including skeletal dysplasia, cardiovascular metabolic diseases, and premature death in old age [3,4]. Additionally, physical fitness also plays an important role in the healthy life of preschool children, such as obesity prevention [5] and determining tibial bone mineral content, structure, and strength in 3–5-year-old children [6]. In addition to health concerns, physical fitness and intellectual maturity have been proven to be linked from an early age, even predicting intellectual maturity in 3–6-year-old children [7] and contributing to successful academic development in youth [8]. These findings highlight the need for promoting physical fitness among children and encouraging them to engage in regular physical activity.

Physical activity has been proven to be one of the important factors promoting physical fitness and is an essential factor of a healthy lifestyle [9,10]. Tan et al. [11] and Wick et al. [12] reported the advantages of physical activity programs over free play in improving the physical fitness of preschool children. The standardized physical activity lessons also exhibited significant advantages over the control group (unorganized physical activity) [13]. In addition, physical activity programs led by kindergartens and teachers have a positive effect on the physical fitness of preschool children [14]. A recent systematic review found that the physical exercise, whether on its own or combined with additional interventions, had beneficial effects on cardiorespiratory fitness, lower-body muscular strength, and speed agility in preschoolers [15]. The formulation of preschool education policy is inclined to using comprehensive exercise and encouraging kindergartens to build their own sports specialties, such as cheerleading, soccer, or basketball, to promote young children’s physical fitness [16]. In conclusion, physical activity plays a crucial role in promoting physical fitness in preschool children. The implementation of structured physical activity programs and the incorporation of exercise into preschool education policies can have a significant effect on the physical fitness and overall health of preschool children.

However, research has indicated that focusing on just one sport can lead to a series of problems in the growth and development of young athletes [17,18]. In addition, it has not been proven whether focusing on only one sport also can lead to problems in the growth and development of preschool children. Studies on the effect of different exercise plans on physical fitness have reported different results because of great differences in the quality and methods used. Moreover, the current evidence does not support a comparison of the effects of different exercise programs, which is not favorable to the selection of physical exercise programs for preschool children. In addition, most previous studies have employed professional coaches or physical educators as the implementers of intervention programs, which limits the generalizability of the findings to guiding physical education practices for preschool children. Studies on teacher-centered physical activity intervention have found no significant advantage over control groups in improving the physical fitness of preschool children [14]. To ensure positive physical fitness development in preschool children, it is important to understand the behavioral attributes and causative mechanisms that promote these outcomes [2].

On that basis, we designed a study to compare different physical exercise programs that have been proven to effectively improve the physical fitness of preschool children and are expected to respond to the evidence gap. Therefore, this study aimed to investigate the effectiveness and differences among these physical exercise programs in improving the physical fitness of preschool children.

## 2. Methods

### 2.1. Study Design and Participants

This study was a single-blind, cluster-RCT study, with the kindergarten class as the cluster for the intervention. The data were sourced from the Physical Exercise on Fundamental Movement Skills and Physical Fitness of preschoolers (PEFP) project [19]. The study population consisted of preschool children aged 4–5 years, who were physically capable of participating in sports and had obtained written consent from their parents or guardians. Participants with severe cognitive or motor impairments were accompanied by a support worker during physical exercise, but were not included in the data collection. Before the end of the interventions, the participants and teachers only participated in the physical exercise of the intervention groups and did not acquire the details of the intervention group allocation. The study was approved by the Ethics Committee of Shanghai Sport University and was registered under the ethical review number 102772019RT034.

In this study, a total of 309 preschool children aged 4 to 5 years were recruited from five kindergartens and cluster randomly assigned to 5 groups: basic movements (BM) group, rhythm activities (RA) group, ball games (BG) group, multiple activities (MA) group, and control (CG) group. The attendance rate of 30% of the participants exceed 4/5 of the total course, and all of the participants completed at least 2/3 of the total course. After preschool children with missing pretest or posttest data were excluded, the final sample consisted of 253 participants (girls 46.3%) with an average age of 4.55 ± 0.28 years: the BG group (n = 55), the RA group (n = 52), the BM group (n = 45), the MA group (n = 44), and the CG group (n = 57). The flow diagram of the research process is shown in Figure 1.

### 2.2. Intervention Procedures

The present study comprised four intervention groups: the BM, RA, BG, and MA groups. Preschool children in the control group participated in unorganized PA, and the details of the interventions have been described elsewhere [19].

The intervention program consisted of structured lessons with a duration of 30 min performed three times a week for 16 weeks. Kindergarten teachers participated in the study and performed the physical exercise interventions after receiving 2 h of training at a local kindergarten. The structure of each lesson consisted of a warm-up period of 5 min, followed by a core exercise period of 20 min and a cool-down activity of 5 min. The study was performed in the winter, and precautions were taken to ensure the safety of the preschool children, such as starting with low-intensity physical activity (e.g., wrist rotations and leg swings), gradually increasing the intensity (e.g., arm rotations and knee-up walk to forceful swinging of arms and on-site running), and then slowly decreasing the intensity. To ensure comparability across the different programs, the core exercise period followed a consistent intensity control principle, whereby every 10 min of sports activities should include at least 5 min of moderate-to-high-intensity physical activity and 2 min of vigorous-intensity physical activity. The interventions were designed as games to increase the children’s interest, with the main differences being in the core exercise content. The interventions were performed within the existing physical activity plans of the kindergartens to avoid additional physical activity for the preschool children in the intervention groups. The intensity of PA was estimated by teachers on the basis of the active behavior of the preschool children and was determined using the “Compendium of Physical Activity” developed by Ainsworth et al. [20] and the Preschool-Age Children’s Physical Activity Questionnaire [21].

Preschool children in the control group participated in unorganized PA. The PA schedules were arranged by the kindergarten without the guidance of teachers, and the types and intensity of activities were determined by the preschool children.

### 2.3. Measurement Procedures

Physical fitness and descriptive data (e.g., age, sex, height, and weight) of preschool children were tested at baseline and at the end of the interventions, and each test was completed within a week. The physical fitness assessment was primarily based on the PREFIT battery, which has demonstrated satisfactory reliability and validity in evaluating the physical fitness of 4–6-year-old children [22].

The physical fitness of the preschool children was evaluated through a comprehensive test battery consisting of measures of cardiorespiratory fitness, musculoskeletal fitness, and motor fitness. The cardiorespiratory fitness of preschool children was assessed by testing the 20 m shuttle run. The musculoskeletal fitness of preschool children was assessed by testing grip strength and standing sit-and-reach. The motor fitness of preschool children was assessed by testing the 10 m shuttle run, balance beam walk, and standing on one foot and hoping. Additionally, anthropometric data, such as height and weight, were collected, and the body mass index (BMI) was calculated from these measurements. The standard testing procedures employed in this study have been described in detail elsewhere [19].

### 2.4. Statistical Analysis

The data were first tested for normality using standardized skewness and kurtosis values. Normally distributed data were presented as the mean and standard deviation, while non-normally distributed data were presented as the interquartile range. One-way analysis of variance (ANOVA) and the Kruskal–Wallis H test were used to examine differences during the pre-experimental stage among groups. The matched samples t-test and Wilcoxon rank-sum test were used to examine the differences of the physical fitness tests in groups before and after intervention. Generalized Linear models (GLMs) were used to assess the differential impacts of the intervention conditions on all outcome indicators for normally distributed data. Generalized Linear mixed models (GLMMs) were used to assess the differential effects of the intervention conditions on all outcome indicators for non-normally distributed data. The intervention condition (CG, BM, RA, BG, and MA) models were adjusted for potential confounders explaining main outcome variance (baseline test results, age, gender, height, weight, and BMI). Bonferroni adjusted pairwise comparisons were employed to analyze differences among conditions, and *p* < 0.05 indicated that the difference is statistically significant. All statistical analyses were performed using SPSS Statistics version 26.0 (IBM Corp, Chicago, IL, USA).

## 3. Results

### 3.1. Participant Characteristics and Physical Fitness Test before Intervention

Table 1 presents participant characteristics and physical fitness tests during the pre-intervention stage. There were significant differences among the groups before the interventions with regard to the balance beam, grip, and 20 m shuttle run test (*p* < 0.05). The scores for balance beam in the BG group were significantly higher than in the other groups (*p* < 0.05). The grip strength of the CG and BG groups was significantly higher than that of the BM and RA groups (*p* < 0.05). The grip strength of the CG group was significantly higher than that of the MA group (*p* < 0.05). The scores for the 20 m shuttle run test in the CG group were significantly higher in the BM and RA groups (*p* < 0.05). The remaining indexes revealed no significant differences among the different groups (Table 1). On the basis of previous literature and results, age, gender, height, weight, and BMI were included as covariates in the subsequent analyses.

### 3.2. Physical Fitness Changes after Intervention

Table 2 presents the results of a matched samples *t*-test and Wilcoxon rank-sum test for the differences between the physical fitness tests before and after the interventions. The pre-post effect sizes exhibited a significant decrease in the sit-and-reach test in all groups after the interventions (*p* < 0.01). In the CG group, the 10 m shuttle run performance of preschool children decreased significantly after the experiment (*p* = 0.009). There were significant improvements in the 20 m shuttle run test (*p* = 0.001), grip (*p* = 0.000), standing on one foot (*p* = 0.027), and skip jump (*p* = 0.009) following the interventions in the BM group. The RA group had significant improvements in the 20 m shuttle run test (*p* = 0.042), grip (*p* = 0.000), and 10 m shuttle run test (*p* = 0.008) after the interventions. The BG group had significant improvements in grip (*p* = 0.000), standing on one foot (*p* = 0.009), 10 m shuttle run test (*p* = 0.000), and skip jump (*p* = 0.002) after the interventions. There was a significant improvement in grip (*p* = 0.000), standing long jump (*p* = 0.000), standing on one foot (*p* = 0.001), 10 m shuttle run test (*p* = 0.000), skip jump (*p* = 0.014), and balance beam (*p* = 0.047) following the interventions in the MA group. The remaining indexes revealed no significant differences before and after the interventions.

Figure 2 presents the results of the generalized linear mixed-model analyses and generalized linear models for each of the physical fitness tests after the interventions. Grip strength was significantly higher in the BG and MA groups than in the BM group (*p <* 0.05), indicating that the BG and MA groups had a significantly better improvement in the grip strength of preschool children than the BM group. The scores for standing long jump were significantly higher in the MA group than in the other groups (*p <* 0.05), indicating that the MA group had a significantly better improvement in the standing long jump of preschool children than the other groups. The scores for the 10 m shuttle run test were significantly lower in the BG and MA groups than in the CG, BM, and RA groups (*p <* 0.05), indicating that the BG and MA groups had a significantly better improvement in the 10 m shuttle run test of preschool children than the CG, BM, and RA groups. The scores for standing on one foot were significantly higher in the BG and MA groups than in the CG and RA groups (*p <* 0.05) and significantly higher in the BM group than in the CG group (*p <* 0.05), indicating that the BG and MA groups had a significantly better improvement in the standing on one foot of preschool children than the CG and RA groups. The scores for skip jump were significantly lower in the BG and MA groups than in the RA group (*p <* 0.05), indicating that the BG and MA groups had a significantly better improvement in the skip jump of preschool children than the RA group. The scores for balance beam were significantly lower in the BG and MA groups than in the RA group (*p <* 0.05) and significantly lower in the BG group than in the BM group (*p <* 0.05), indicating that the BG and MA groups had a significantly better improvement in the balance beam of preschool children than the RA group. However, the scores for the 20 m shuttle run test and the sit-and-reach test revealed no significant differences among the different groups.

## 4. Discussion

Preschool children undergo a period of rapid physical growth and maturation of the nervous system, requiring the development of corresponding physical fitness, such as agility, strength, and reaction speed [23,24]. Evidence from systematic reviews focuses on the strong association between cardiorespiratory fitness and musculoskeletal fitness and the development of motor competence throughout early years, childhood, and adolescence, with increasing strength with age [2,25]. Based on this evidence, it is rational to believe that the importance of physical fitness in preschool children should be the same as that of older children [26]. The present study aimed to identify more effective physical exercise programs to improve the physical fitness of preschool children and provide evidence for the implementation of preschool physical education. Following these 16-week interventions in preschool, children exhibited improvements in all physical fitness tests after intervention for all intervention groups, except for the sit-and-reach test, and the balance beam test in the RA group. In the CG group, the preschool children showed no significant increase in all physical fitness indicators of preschool children. The BG and MA groups had a certain advantage over the BM, RA, and CG groups in improving the physical fitness of preschool children.

In terms of cardiorespiratory fitness, pre-post effect sizes exhibited significant improvements in the 20 m shuttle run test in the BM and RA groups, which is consistent with previous studies [15]. However, the improvement in the 20 m shuttle run test before and after the interventions in the BG and MA groups was not as pronounced and not statistically significant. This may be because the baseline cardiorespiratory fitness levels of the preschool children in the BM and RA groups were lower than those of the children in the BG and MA groups. Previous research has indicated that the baseline level of physical fitness in preschool children can affect the effect size of interventions, with higher baseline scores leading to smaller improvements and lower baseline scores leading to larger changes [27,28]. In addition, after the baseline test value and other confounding factors were adjusted, the BG and MA groups demonstrated an advantage in terms of improving cardiorespiratory fitness when compared with the BM and RA groups. Systematic review and meta-analysis results from recent studies have indicated that all types of physical activity programs, including free play, can improve the cardiorespiratory fitness of preschool children to a certain extent [15,29], which is consistent with the findings of this study.

The muscle strength (grip and standing long jump) of preschool children in all intervention groups, including the control group, obviously improved after the interventions, which is consistent with previous research findings [15,27]. The BG and MA groups demonstrated advantages over the BM and RA groups in terms of grip strength improvement, whereas the MA group demonstrated significant improvements in standing long jump performance when compared with the other groups. However, the flexibility (sit-and-reach) of preschool children in all intervention groups and the control group decreased significantly, which contrasts with previous findings [15,27]. Long-term studies have indicated that preschool children’s physical fitness will gradually increase with age [30,31], except for flexibility, which may exhibit little change or even decrease without targeted practice [27,32]. In addition, the decline of the preschool children’s flexibility may be affected by the season (children’s clothing and temperature). The baseline test was in the autumn, when clothes and temperature had little effect on children’s motor performance. The interventions ended in the winter, when cold temperatures and heavy clothes have a great effects on children’s motor performance [33]. It is known that possessing adequate flexibility, range of motion, and muscle strength can mitigate the risk of injury in sports or everyday activities, particularly in later life, when the negative effect of decreased flexibility on health cannot be disregarded [34]. Therefore, the flexibility exercise of preschool children should be an important part of the physical exercise program formulation. The results of this study suggest that the MA intervention exhibited advantages in improving the muscle strength of preschool children when compared with other physical exercise programs and the control group. However, further research is warranted to better understand the effect of various physical exercise programs on the flexibility of preschool children.

The motor fitness of preschoolers was obviously improved in all intervention groups after the 16-week interventions, and the intervention groups had certain advantages over the CG group. Previous research, and systematic review and meta-analysis also, indicated that the designed physical activity programs had a positive effect on the motor fitness of preschool children [11,15,29], similar to the results of this study. The BG and MA groups displayed obvious advantages in improving the motor fitness of preschool children when compared with the BM, RA, and CG groups. This may be caused by a close relationship between the performance of children’s motor fitness and the level of motor skills [2]. A study on the effect of different exercise programs on the motor skills of preschool children has indicated that multilateral exercise has certain advantages over specific programs of rhythmic gymnastics and soccer [18]. The physical exercise of the MA group may better improve the motor performance of preschool children by better improving their motor skills. In this study, ball games have similar intervention effects on preschoolers’ motor fitness with multiple activities. In addition, research has indicated that the motor fitness of preschool children was significantly improved with small improvements in cardiovascular fitness. The BG and MA groups exhibited advantages in improving cardiorespiratory fitness when compared with the BM, RA, and CG groups. This may help explain why BG and MA can better improve the motor fitness of preschool children.

In summary, the MA group had advantages over the BM, RA, and CG groups in terms of the improvement of the physical fitness of preschool children. In addition, in this study, the BM and RA groups had no advantages over the CG group with regard to improvement of cardiorespiratory fitness, musculoskeletal fitness, and motor fitness. These results are similar to those of another study that found that teacher-centered intervention granted preschool children no advantage over the control group in terms of motor fitness [14]. There is an evidence gap with regard to the effect of different physical exercise programs on the physical fitness of preschool children, and there were no similar results for reference to verify whether the results of this study are reasonable. However, relevant studies have indicated that early specialized sports training or focusing on the development of just one sport may lead to a series of growth and development problems, such as physical and physiological imbalance, unilateral muscle development, risk of injury, coordinated development disorder, and limitations on differentiated skill acquisition, and even a negative effect on mental health, and can also reduce children’s enthusiasm for PA participation [17,35]. In addition, studies have indicated that the diversified sports activity module and the structured multisport program have significant advantages over free play or conventional sports activity in improving the physical fitness of preschool children [27,36]. According to the research of Stodden et al. [37] and Lubans et al. [38], PA, physical fitness, and motor skills all reinforce each other, and multilateral exercise has certain advantages over the single exercise mode in improving the motor skills of preschool children [17]. This evidence can help explain why multiple activity programs can better improve the physical fitness of preschool children.

There are several limitations that need to be addressed in this study. The first is in terms of sample representation; because of the scale and difficulty of the experiment, only 4–5-year-old preschoolers were included in this study. Therefore, the results of this study may not be applicable to all preschool children. Second, the baseline level of physical fitness in the experimental groups was not balanced. The improvement after the interventions will be greater if the baseline test level is low. In addition, the physical environments of the baseline test and post-intervention test were relatively different. Therefore, the significance of analyzing the improvement of physical fitness before and after the interventions is limited. However, we used a mixed-effects model to adjust the effect of the baseline test results, gender, and other factors in the intervention effect. In addition, all kindergartens participating in the experiment were in the same community, and the test environment was similar. Finally, the number of preschool children in each group included in the analysis was not balanced, but the minimum sample size that meets the statistical analysis was 30 children per group [19].

## 5. Conclusions

Physical exercise programs designed for preschool physical education have positive effects on the physical fitness of preschool children. Compared with the exercise programs with a single project and action form, comprehensive exercise programs with multiple action forms can better improve the physical fitness of preschool children.

## Figures and Tables

**Figure 1 ijerph-20-04254-f001:**
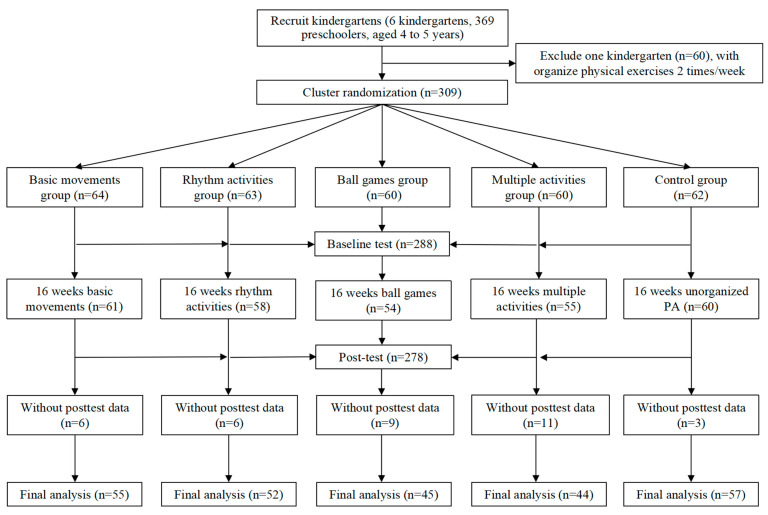
Flow chart of research method.

**Figure 2 ijerph-20-04254-f002:**
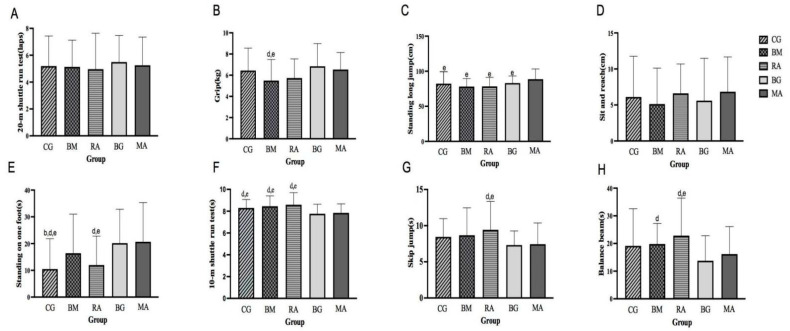
The results of the generalized linear mixed-model analyses for physical fitness tests after intervention. CG, Control group; BM, Basic movements group; RA, Rhythm activities group; BG, Ball games group; MA, Multiple activities group. The 20 m shuttle run test (**A**), Grip (**B**), Standing long jump (**C**), Sit-and-reach (**D**), Standing on one foot (**E**), 10 m shuttle run test (**F**), Skip jump (**G**), Balance beam (**H**) in CG, BM, RA, BG, and MA groups. b: CG vs. BM *p* < 0.05, d: CG, BM, RA vs. BG *p* < 0.05, e: CG, BM, RA, BG vs. MA *p* < 0.05, adjusted for age, sex, BMI, height, weight, and physical fitness test in the pre-experiment stage under each condition.

**Table 1 ijerph-20-04254-t001:** Participant characteristics and Physical Fitness Test before each intervention.

Variables	CG	BM	RA	BG	MA	*p*
Age (years)	4.57 (0.59)	4.53 (0.54)	4.53 (0.48)	4.54 ± 0.26	4.54 ± 0.27	0.92
Gender (boys/girls)	34/23	30/25	32/20	20/25	20/24	0.30
Height (cm)	110.86 ± 5.57	109.90 (4.00)	108.40 (3.60)	111.34 ± 5.56	109.67 ± 5.15	0.06
Weight (kg)	18.75 (3.30)	18.00 (2.65)	18.35 (2.60)	18.50 (3.67)	18.34 ± 2.30	0.19
BMI (kg/m^2^)	15.57 ± 1.64	14.97 (1.65)	15.41 (2.11)	15.69 ± 1.63	15.22 ± 1.28	0.32
20 m shuttle run test (laps)	5.40 ± 2.16	5.00 (5.00) ^a^	4.00 (4.00) ^a^	5.27 ± 2.00	4.86 ± 2.58	0.01
Grip (kg)	5.88 ± 2.21	4.27 ± 1.69 ^a,d^	4.33 ± 1.98 ^a,d^	5.39 ± 1.96	4.70 ± 1.63 ^a^	<0.01
Standing long jump (cm)	79.49 ± 16.33	75.64 ± 13.72	77.21 ± 12.08	81.65 ± 13.51	78.27 ± 13.57	0.33
Sit-and-reach (cm)	8.74 ± 5.15	6.65 ± 5.67	9.05 ± 4.90	8.50 (5.20)	9.13 ± 4.14	0.17
10 m shuttle run test (s)	8.68 ± 0.93	8.68 ± 1.22	8.95 ± 0.99	8.53 ± 1.03	8.34 (1.73)	0.30
Skip jump (s)	8.37 (3.69)	8.40 (4.32)	9.07 (4.79)	7.84 (4.17)	7.49 (4.28)	0.50
Balance beam (s)	18.63 (14.63) ^d^	18.91 (12.27) ^d^	18.93 (14.08) ^d^	12.8 (11.30)	17.35 (46.46) ^d^	0.03
Standing on one foot (s)	8.31 (9.44)	7.40 (11.70)	7.67 (10.90)	8.91 (10.86)	10.11 (12.64)	0.43

Normally distributed data are represented by mean ± standard deviation (Mean ± SD), and non-normal distribution data are represented by the median and interquartile range. *p*-values were generated using one-way ANOVA and Kruskal–Wallis H test. CG, Control group; BM, Basic movements group; RA, Rhythm activities group; BG, Ball games group; MA, Multiple activities group. ^a^: BM, RA, BG, MA vs. CG *p* < 0.05; ^d^: CG, BM, RA, MA vs. BG *p* < 0.05.

**Table 2 ijerph-20-04254-t002:** Difference of Physical Fitness before and after intervention.

Variables	CG	BM	RA	BG	MA
20 m shuttle run test (laps)	0.00 (2.00)	1.00 (2.00) **	0.00 (3.00) *	0.22 ± 1.99	0.00 (2.75)
Grip (kg)	0.57 ± 1.99	1.21 ± 2.12 **	1.40 ± 2.20 **	1.45 ± 2.10 **	1.83 ± 1.86 **
Standing long jump (cm)	2.71 ± 13.40	2.47 ± 11.93	1.17 ± 12.41	1.42 ± 12.19	10.34 ± 12.57 **
Sit-and-reach (cm)	−2.50 (4.45) **	−2.30 (5.70) **	−2.44 ± 4.55 **	−3.10 (4.95) **	−2.30 ± 3.89 **
10 m shuttle run test (s)	0.38 ± 1.08 **	−0.23 ± 1.21	−0.35 ± 1.10 **	−0.78 ± 1.22 **	−0.77 (1.03) **
Skip jump (s)	−0.31 (4.01)	−0.84 (3.16) **	−0.33 (3.02)	−1.61 ± 3.27 **	−0.93 (3.30) *
Balance beam (s)	−1.75 (12.58)	−0.81 ± 8.73	0.27 (14.95)	−2.03 ± 10.35	−2.50 (15.15) *
Standing on one foot (s)	−0.52 (8.34)	2.87 (10.78) *	2.05 (7.37)	6.73 ± 16.57 **	6.19 ± 11.16 **

Normally distributed data are represented by mean ± standard deviation (Mean ± SD), matched samples *t*-test and non-normal distribution data are represented by the median and interquartile range. *p*-values were generated using one-way ANOVA and Kruskal–Wallis H test. CG, Control group; BM, Basic movements group; RA, Rhythm activities group; BG, Ball games group; MA, Multiple activities group. *: Pre-Intervention vs. Post-Intervention *p* < 0.05, **: Pre-Intervention vs. Post-Intervention *p* < 0.01.

## Data Availability

Data is unavailable due to privacy and ethical restrictions.

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
