# Peer review of "The Effect of Different Physical Exercise Programs on Physical Fitness among Preschool Children: A Cluster-Randomized Controlled Trial"

_ijerph, 2023, doi:10.3390/ijerph20054254_

Round 1

Reviewer 1 Report

Even though the authors are mentioning tables and figures in the main text, I can not see them. Please provide tables and figures as the results are tough to understand this way. 

Also, supplementary materials should be removed as they are not in the English language and thus are not understandable to most readers.

Author Response

  1. Even though the authors are mentioning tables and figures in the main text, I can not see them. Please provide tables and figures as the results are tough to understand this way.

Response: Thanks for your suggestion. The figures and tables of the manuscript were uploaded independently when the manuscript were submitted, and now the figure and table have been put into the manuscript.

  1. Also, supplementary materials should be removed as they are not in the English language and thus are not understandable to most readers.

Response: Thanks for your suggestion. The supplementary materials of the manuscript was processed by the review system. We will not upload supplementary materials when uploading revised manuscript.

  1. Did you perform a pilot study? How did you calculate the sample size?

Response: Thanks for your suggestion. We conducted a pilot study prior to the formal intervention, but the pilot study was only conducted for 20 days due to the COVID-19 outbreak control. Through the pilot study we verified the feasibility of the physical exercise programs and adapted the physical exercise content according to the implementation of the physical exercise program.

Prior to the conduct of this study we had published the study protocol, in which the estimation of the sample size was described in detail, with a minimum sample size of 30 preschool children per group.

  1. Which criteria did you use to choose specific kindergarten groups?

Response: Thanks for your suggestion. The criteria for selecting kindergartens were: 1) in the same community, 2) similar size of kindergartens with similar number and type of physical activity facilities, and 3) similar outdoor physical activity schedules of kindergartens.

  1. Were study participants taking any medications or having chronic conditions? Please elaborate on attendance rate for physical activity classes. Were participants enrolled in some other (afternoon) activities?

Response: Thanks for your suggestion. Our experiment was conducted in Putuo District, Shanghai, China, where the kindergarten has a resident doctor who assesses the physical condition of the students. And the kindergarten organizes annual participation of preschoolers in formal medical examinations in the hospital.

We supplemented our methodology with the overall attendance of the students. “The attendance rate of 30% of the participants exceed 4/5 of the total course, and all of the participants exceed 2/3 of the total course.” Please check lines 124-125.

All kindergartens participating in the experiment had similar physical activity schedules. All performed twice physical activities per day, approximately 90-min in the morning and 45-min in the afternoon. The intervention in this study took only 30-min in the morning, and the rest of the physical activity schedule remained unchanged.

  1. Results are presented in awkward way: there are no information on baseline physical fitness and why there are differences among some groups (also, in Table 1 please provide post-hoc analyses and explain the observed differences). Furthermore, I would suggest that results in Table 2 are shown as both pre- and post-activity values, not only the differences. Also, please provide exact P values. I don’t see the point of data presentation in the form of the Figure 2, since there were some differences in baseline (pre-exercise) levels of some fitness variables, age, gender, and size of the body, which is not visible in Figure 2.

Response: Thanks for your suggestion. The baseline data have been shown in Table 1. There are many indicators and subgroups for which presenting pre- and post-intervention data in the table was not conducive to comparison. Some indicators differed at baseline, so pre- and post-intervention effect sizes were used for greater clarity; a mixed model was also used to adjust for factors that differed at baseline as confounders and to compare changes in indicators before and after the intervention.

  1. The discussion has no logical flow and should be restructured. Furthermore, the

manuscript would strongly benefit from language corrections.

Response: Thanks for your suggestion. We have reconceived and restructured the discussion section, please check the discussion section in the manuscript.

  1. We have reconceived and restructured the discussion section, please check the discussion section in the manuscript.What was the girls’ percentage (there are different numbers in abstract and methods)?

Response: Thanks for your suggestion. The percentage of girls in the abstract is pre-experimental and the percentage of girls in the methods is for the final analysis sample. We have made adjustments to face create a similar misunderstanding. Please check lines 31-33.

Reviewer 2 Report

This paper examines the effectiveness and the differences between different physical exercise programs in improving preschool 18 children’s physical fitness.   309 preschool children (girl 49.8%) aged 4-5 years participated in the experiment, recruited from 5 kindergartens and were cluster-randomly allocated to 5 groups, Basic movements (BM) group, Rhythm activities (RA) group, Ball games (BG) group, Multiple activities (MA) group, or Control (CG)group.  Intervention groups received designed physical 22 exercise programs three times per week lasting 30 minutes for 16 weeks.  Results indicated that physical exercise programs designed for preschool physical education have positive effects on the physical fitness of preschool children.  Compared with the exercise programs with single project and action form, the comprehensive exercise programs with multiple action forms can better improve the physical fitness of preschool children.

Title: Should state the nationality/country where the study was conducted.

Abstract and Introduction: I suggest proofreading from a native English speaker to improve the flow, spelling and grammatical errors but also the lack of paragraphs in the introduction.

Methods: Same as above.  Difficult to follow.

Results: Missing p-values.  Missing table 1 and table 2.  There should be at least graphs representing the results.

Discussion: Lack of structure and difficult to follow.

I suggest the others proofread the paper with the help of an English proofreader as it is difficult to follow, make the suggested additions and resubmit.

Author Response

  1. Title: Should state the nationality/country where the study was conducted.

Response: Thanks for your suggestion. Despite being conducted in Shanghai, China, 1) the physical activities for the intervention were selected from physical activity approaches for young children commonly used in international studies, so the intervention should be generalisable across countries and regions. 2) no regional differences in preschoolers' response to physical activity have been reported in relevant studies. Therefore, the intervention outcomes produced by the intervention subjects receiving these interventions should be  generalisable. 3) The manuscript states the region in which the intervention was delivered. Therefore, after consideration we decided not to emphasise the country in the title.

  1. Abstract and Introduction: I suggest proofreading from a native English speaker to improve the flow, spelling and grammatical errors but also the lack of paragraphs in the introduction.

Response: Thanks for your suggestion. We have professionally edited the language in the manuscript.

  1. Methods: Same as above. Difficult to follow.

Response: Thanks for your suggestion. We have professionally edited the language in the manuscript.

  1. Results: Missing p-values.Missing table 1 and table 2. There should be at least graphs representing the results.

Response: Thanks for your suggestion. The figures and tables of the manuscript were uploaded independently when the manuscript were submitted, and now the figure and table have been put into the manuscript.

  1. Discussion: Lack of structure and difficult to follow.

Response: Thanks for your suggestion. We have reconceived and restructured the discussion section, please check the discussion section in the manuscript.

  1. I suggest the others proofread the paper with the help of an English proofreader as it is difficult to follow, make the suggested additions and resubmit.

Response: Thanks for your suggestion. We have reconceived and restructured the manuscript, and professionally edited for language in the manuscript.

Reviewer 3 Report

Thank you for allowing me to evaluate your manuscript.

I acknowledge th eauthors' effort to capture a large sample and to perform a mid-term intervention, but there are several aspects that I would like to comment on.

The objective they set was to evaluate the effect of different physical exercise programs on the physical condition of preschool-age children.

I believe that, as the authors comment in the introduction, it has already been reported in other studies that physical exercise, combined or not with an additional intervention, had beneficial effects on cardiorespiratory health. It is also known that concentrating on a single sport can lead to a number of growth problems, and that multiple activities are more beneficial. So the conclusions obtained in the study are highly predictable and provide little new scientific evidence.

On the other hand, I consider that the particular characteristics of the age of the sample mean that the results cannot be directly attributed to the physical activity programs carried out. Children from 4 to 5 years old are very active in their daily lives, and I believe that it is not possible to associate the results obtained only with the proposed activities alone.

Moreover, the limitations described by the authors are of sufficient weight to be taken into account and see their direct impact on the quality of the results obtained.

Having said this, and focusing on the structure of the article, I think that the presentation of the results obtained in the study should be rethought. They are written in a confusing way and must be accompanied by a graphic or image to help interpret them better

As for the discussion,  it should be rewritten, since it hardly justifies the relationship between the type of program carried out and the physical improvements obtained.

Author Response

  1. I believe that, as the authors comment in the introduction, it has already been reported in other studies that physical exercise, combined or not with an additional intervention, had beneficial effects on cardiorespiratory health. It is also known that concentrating on a single sport can lead to a number of growth problems, and that multiple activities are more beneficial. So the conclusions obtained in the study are highly predictable and provide little new scientific evidence.

Response: Thanks for your suggestion. There are two contributions of this study. First, we demonstrate that a multi-modal physical activity program in the early childhood years is better for developing physical fitness. Previous studies have demonstrated the advantages of physical activity over free play in improving the physical health of preschool children. We do not have more information on other countries, but in China we found that more than 90% of kindergartens only do free play, and a small number of kindergartens do a single physical activity such as rhythmic gymnastics, basketball, soccer, or only simple basic movement exercises (results from the cross-sectional survey part of which research project of this study). Therefore, there is a need for such research findings to support policy development in China. Second, although previous studies have demonstrated that structured physical activity improves the physical health of young children more than free play, the intervention implementers are mostly experienced coaches or physical educators. Few studies have been led by kindergarten teachers, leading to questions about the application value of their findings. The physical activities in this study fully considered the opinions of kindergarten teachers and the intervention was implemented by kindergarten teachers, which greatly improved the practical value of the findings.

  1. On the other hand, I consider that the particular characteristics of the age of the sample mean that the results cannot be directly attributed to the physical activity programs carried out. Children from 4 to 5 years old are very active in their daily lives, and I believe that it is not possible to associate the results obtained only with the proposed activities alone.

Response: Thanks for your suggestion. Your concerns have been taken into our consideration, therefore, we have set up a control group to reflect the effect of different physical exercise programs on the physical health of children aged 4-5 years.

  1. Moreover, the limitations described by the authors are of sufficient weight to be taken into account and see their direct impact on the quality of the results obtained.

Response: Thanks for your suggestion. Regarding the limitations of the study, we acknowledge that our study is not perfect. However, we all know that a study can only address a specific research question as best as possible and minimize the influence of other factors on the findings.

  1. Having said this, and focusing on the structure of the article, I think that the presentation of the results obtained in the study should be rethought. They are written in a confusing way and must be accompanied by a graphic or image to help interpret them better.

Response: Thanks for your suggestion. We have professionally edited the language, and reconceived and restructured the manuscript. The figures and tables of the manuscript were uploaded independently when the manuscript were submitted, and now the figures and tables have been put into the manuscript.

  1. As for the discussion, it should be rewritten, since it hardly justifies the relationship between the type of program carried out and the physical improvements obtained.

Response: Thanks for your suggestion. We have reconceived and restructured the discussion section, please check the discussion section in the manuscript.

Round 2

Reviewer 1 Report

I have no further comments.

Author Response

Thank you very much for your valuable time and suggestions on our manuscript.

Reviewer 2 Report

Dear authors,

1. The text needs extensive proofreading by a native English speaker.  There are quite a few mistakes in the text, i.e. researches instead of researchers etc.  This could have been picked if the proofreading had been done by a professional.

2. Methods: Authors keep referring to a non-parametric test, but they fail to include the appropriate name of the test. No indication of a p-value for statistical significance.

3. Results: Table 2 included does not indicate where there is a statistical significance.  Authors should indicate whether they talk about statistical or clinical significance.  The quality of the figures must be improved.

Author Response

  1. The text needs extensive proofreading by a native English speaker.  There are quite a few mistakes in the text, i.e. researches instead of researchers etc.  This could have been picked if the proofreading had been done by a professional.

Response: Thank you for your patience and suggestions. The manuscript has been extensive proofreading by a native English speaker and the Certificate of English Language Editing has been uploaded with the manuscript. We still cannot guarantee that the current language is completely appropriate, as we are not native English speakers. But we have done all that we can do.

  1. Methods: Authors keep referring to a non-parametric test, but they fail to include the appropriate name of the test. No indication of a p-value for statistical significance.

Response: Thanks for your suggestion. We have added the names of the non-parametric tests that we used and statistical significance for p-values into the methods section. “One-Way analysis of variance (ANOVA) and Kruskal-Wallis H test were used to examine differences during the pre-experimental stage among groups. Matched samples t-test and Wilcoxon rank sum test were used to examine differences of the physical fitness tests in groups before and after intervention. And P < 0.05 indicated that the difference is statistically significant.” Please check lines 179-183, 190-191.

  1. Results: Table 2 included does not indicate where there is a statistical significance.  Authors should indicate whether they talk about statistical or clinical significance.  The quality of the figures must be improved.

Response: Thanks for your suggestion. We have enhanced the description of the analysis results, such as 1) We have supplemented the meaning of the P value in Table 2. Please check lines 248-252. 2) We have supplemented the clinical significance of the statistical results in Figure 2. Please check lines 253-276. Table 2 shows the analysis of changes in physical fitness of all experimental groups before and after the interventions. Whether there is a * sign has been used as a symbol of whether the change value is statistically significant. The comparison results of physical fitness improvement among the experimental groups are shown in Figure 2.

Reviewer 3 Report

I believe that the changes that have been made are correct and have greatly improved the clarity and exposition of the article.

Author Response

(The authors gave the same response as above.)
